

# Soil characteristic changes and quality evaluation of degraded desert steppe in arid windy sandy areas

Jing Ma[1,2,*], Jianrong Qin[3,*], Hongbin Ma[1], Yao Zhou[1], Yan Shen[1], Yingzhong Xie[1] and Dongmei Xu[1]

[1] School of Agricultural, Ningxia University, Yinchuan, China
[2] Agricultural Economy and Information Technology Research Institute of Ningxia Academy of Agricultural and Forestry Sciences, Yinchuan, China
[3] Chengdu Park City Construction & Development Research Institute, Chengdu, China
* These authors contributed equally to this work.

## ABSTRACT

Grassland degradation has become a serious problem in some areas, making it necessary to quantitatively evaluate this process and its related factors. The study area was the arid windy sandy area in eastern Ningxia. The purpose of this study was to explore how soil properties and quality change during the process of grassland degradation in arid windy sandy areas. We looked at undegraded, lightly degraded, moderately degraded, and severely degraded desert steppe to study the physical, chemical, and biological changes at 0–5 cm, 5–15 cm, and 15–30 cm soil depths at different degradation degrees. We also analyzed the correlations across soil factors, established the minimum data set, and used the soil quality index (SQI) to evaluate the soil quality of grassland at different degradation degrees. The results showed that with grassland degradation, the soil bulk density increased; the soil clay, moisture, organic matter, total nitrogen, and available potassium content decreased; and the number of soil bacteria, actinomycetes, and fungi, as well as the activity of urease, polyphenol oxidase, protease, phosphatase, and sucrase, decreased. As soil depth increased, soil bulk density increased; the soil moisture, organic matter, available potassium, and available phosphorus content decreased; and soil microorganisms accumulated in the upper soil of undegraded, lightly, and moderately degraded grassland. There was also a positive correlation among the soil clay content, moisture content, organic matter content, total nitrogen content, available potassium content, microorganism quantity, and enzyme activity, while soil bulk density was negatively correlated with the above factors. The minimum data set for the soil quality evaluation of the degraded desert steppe was comprised of soil organic matter content, soil total nitrogen content, soil available phosphorus content, and phosphatase activity. Based on the minimum data set, we calculated the SQI of the grassland at different degradation degrees and found that the ranking based on overall soil quality was undegraded >lightly degraded >moderately degraded >severely degraded grassland. The results showed that the degradation of desert steppe in arid windy sandy areas had relatively consistent effects on the physical, chemical, and biological traits of the soil. The minimum data set can be used to replace the total data set when evaluating the soil quality of the desert steppe at different degrees of degradation.

Corresponding author
Hongbin Ma, ma_hb@nxu.edu.cn

# INTRODUCTION

Grassland degradation has been more widely observed in recent years, and has become an increasingly serious problem in certain areas (*Abdalla et al., 2018*; *Harris, 2010*; *Liu et al., 2018*; *Robinson, Li & Hou, 2017*; *Schonbach et al., 2011*; *Wei et al., 2020*; *Zhang et al., 2014*). Grassland degradation is defined as the process of environmental degradation and desertification caused by overexploitation, poor management, climate warming, and drought (*Li et al., 2011*). It is the overall decline of grassland productivity caused by human activities and natural processes (*Belayneh & Tessema, 2017*; *Tiscornia, Jaurena & Baethgen, 2019*; *Yao et al., 2016*; *Zhang et al., 2018*). Grassland degradation reduces vegetation coverage and species diversity, changes species composition, and reduces stored organic carbon and nutrients in soil (*Gao et al., 2019*; *Han et al., 2018*; *Liu, Schleuss & Kuzyakov, 2017*; *Mchunu & Chaplot, 2012*). It also has adverse effects on the productivity of terrestrial ecosystems and the balance of the regional and ecological environment (*Bai et al., 2008*; *Nacun et al., 2018*; *Shen, Liu & Zhou, 2015*; *Xu et al., 2020*). It is necessary to further quantitatively evaluate grassland degradation and its related factors, which have been subjects of intense research in recent years.

Soil is a complex biological system that plays a key role in plant growth, organic matter decomposition, nutrient cycling, and water retention (*Ritz et al., 2009*; *Saglam, Dengiz & Saygin, 2015*; *Zhou et al., 2019*). Against the backdrop of global grassland degradation, changes in soil properties in degraded grassland have aroused widespread concern. The most direct manifestation of grassland degradation is the reduction of vegetation cover (*Babel et al., 2014*; *Dlamini et al., 2014*), which also increases the deterioration of soil physical properties and the infertility of nutrients. Therefore, the core problem of grassland degradation is soil degradation. Soil degradation weakens the soil structure, reduces organic matter content, intensifies soil desertification, and reduces soil productivity (*Ma, Wang & Shen, 2020*). Although grassland degradation is rapidly expanding, quantitative information about the impact of grassland degradation on soil properties is still largely unavailable (*Yuan et al., 2019*).

Soil quality is a key aspect of ecosystem function and agricultural sustainability, and it reflects abiotic and biotic interactions in the process of maintaining plant and animal productivity (*Nabiollahi et al., 2018*; *Nosrati & Collins, 2019*; *Rahmanipour et al., 2014*; *Sione et al., 2017*). Monitoring soil quality is necessary to assess changes of soil properties and judge whether soil is improving or degrading (*Bilgili, Kucuk & Van Es, 2017*; *Santos-Frances et al., 2019*). Soil quality is a comprehensive reflection of its physical, chemical, and biological characteristics (*Maurya et al., 2020*; *Zuber et al., 2017*). There is no single measurement method that is used to directly determine soil quality, but it is instead assessed by measuring soil physical, chemical, and biological characteristics (*Imamoglu & Dengiz, 2019*; *Yu et al., 2018*). Soil quality assessment is generally performed

by selecting a set of soil characteristics that are considered soil quality indicators (*Guo et al., 2017*; *Vasu et al., 2016*). Soil organic matter is one of the most important indicators used to measure soil fertility (*Sharma et al., 2016*). Nitrogen is a major nutrient for vegetation growth and an important indicator for evaluating soil quality (*Pham, Nguyen & Kappas, 2018*). Soil microorganisms and enzyme activities are early indicators for assessing the degree of soil degradation, and are sensitive to soil disturbances related to nutrient cycling and organic matter dynamics (*Guo et al., 2018*; *Nosrati & Collins, 2019*; *Pandey, Agrawal & Bohra, 2015*). There are many methods used for evaluating soil quality: the comprehensive index method (*Zhang et al., 2017*), fuzzy comprehensive evaluation method (*Wang, Yang & Shan, 2001*), principal component analysis method (*Juhos et al., 2019*), gray correlation method (*Li, Tang & Li, 2004*), and the soil quality index (SQI) (*Zhou et al., 2020*). Soil quality evaluations are considered important and the soil quality evaluation of grassland ecosystems has attracted increasing attention. However, a relatively comprehensive soil quality evaluation system for grasslands has not been completed, and there have been few studies on the soil quality evaluations of desert grassland.

Simple to calculate and easy to use, the soil quality index (SQI) is an important tool used for evaluating and quantifying soil quality (*Nabiollahi et al., 2017*; *Zhao et al., 2019*). Soil quality indexing involves three steps: selecting appropriate indicators and determining the minimum data set (MDS), converting the indicators and assigning the weights, and integrating all indicator scores into an SQI (*Nabiollahi et al., 2017*). The total data set (TDS) and MDS have been widely used for soil quality assessments (*Zhou et al., 2020*). The TDS contains all soil quality indicators, while the MDS contains fewer but more important indicators. Using the MDS can reduce the workload of data measurement and analysis because the most important indicators containing sufficient information for quality evaluation are selected (*Jahany & Rezapour, 2020*). When establishing the MDS, indicators can be selected based on expert opinions or statistical processes, such as principal component analysis (*Askari & Holden, 2014*; *Juhos et al., 2019*; *Lima et al., 2013*). The soil quality evaluation process can be affected when only physical and chemical indicators are selected but more sensitive biological indicators are ignored. A data set that contains at least one physical, chemical, and biological indicator can more accurately reflect the soil quality (*Zhou et al., 2020*).

The arid windy sandy area of eastern Ningxia is located on the southern edge of the Mu Us Sandy Land. The zonal vegetation is a desert steppe. Arid and rainless with strong winds, the sand-covered ecosystem is extremely fragile. It is an important ecological barrier, animal husbandry base, one of the largest areas of desertified land, as well as the area in Ningxia most severely damaged by sandstorms. Due to years of overgrazing coupled with harsh natural conditions, the vegetation has been degraded, the soil has deteriorated, and the health of the ecosystem has been seriously threatened.

To investigate how soil properties and quality change during the process of grassland degradation in arid windy sandy areas, we used the desert steppe in the arid windy sandy area of eastern Ningxia in our research. This study discussed the variations in the physical, chemical, and biological characteristics of soil in the grassland at different degrees of degradation, analyzed the correlations among soil factors, and evaluated the

comprehensive condition of soil quality with the help of the MDS and SQI. Our results provide a basis for ecological restoration and scientific management of the desert steppe.

## MATERIALS AND METHODS

### Site description

The study area was located in the middle-north region of Yanchi County, eastern Ningxia (37°44′–38°10′N, 106°50′–107°40′E), a typical arid windy sandy area connected to the Mu Us Sandy Land to the north and the Loess Plateau to the south. The terrain transitions from Ordos' gently sloping hills in the north to loess hills in the south, and the climate is typical mid-temperate semiarid continental. The average annual temperature is 8.1 °C. The annual precipitation is 250–350 mm. The annual evaporation is 2,403.7 mm. The climate is dry with little rain and strong winds during all four seasons. It is hot in the summer, cold in winter, and windy in spring. Drought and sandstorms often occur. The zonal soil is light sierozem, and the zonal vegetation is desert steppe mainly composed of xerophytes and mesophytes, such as *Artemisia ordosica*, *Pennisetum centrasiaticum*, *Glycyrrhiza uralensis*, *Stipa breviflora*, *Sophora alopecuroides*, and *Lespedeza potaninii*. The soil is poor and the vegetation is sparse. Due to drought, lack of rain, strong wind erosion, long-term overgrazing, reclamation, and other negative developments, the desert steppe in this area has been seriously degraded. Since grazing exclusion was enacted in 2003, the grassland ecological environment has improved significantly.

### Study methods

#### Plot setting

The study area and sample sites are shown in Fig. 1. The research method of substituting space for time is commonly-used (*Liang et al., 2002*; *Shen et al., 2015*; *Zhao et al., 2020*). We selected grasslands with consistent terrain and soil conditions, but different degrees of degradation, and set up 54 sample plots. In each sample plot, five quadrats were set at equal intervals in a diagonal direction for vegetation investigation with 1 m × 1 m of herbaceous plants and 10 m × 10 m of shrubs. Five points were arranged in an "S" shape to determine soil properties and for soil sample collection. Based on the vegetation survey and measurement data, we used the cluster analysis method (*Zhang, Li & Xie, 2008*) to classify the degradation degrees of the desert steppe in the study area into undegraded (UD), lightly degraded (LD), moderately degraded (MD), and severely degraded (SD) grasslands. There were seven undegraded grasslands, 39 lightly degraded grasslands, three moderately degraded grasslands, and five severely degraded grasslands. The vegetation profiles of the sample plots at each degradation degree are shown in Table 1.

#### Measuring items and methods

(i) Vegetation investigation: We studied the vegetation during the period of vigorous vegetation growth in August once a year across 2 years. The geographic coordinates and elevations of each sample plot were recorded during the survey. The species composition in each plot was counted, and the natural heights of 30 plants of each species

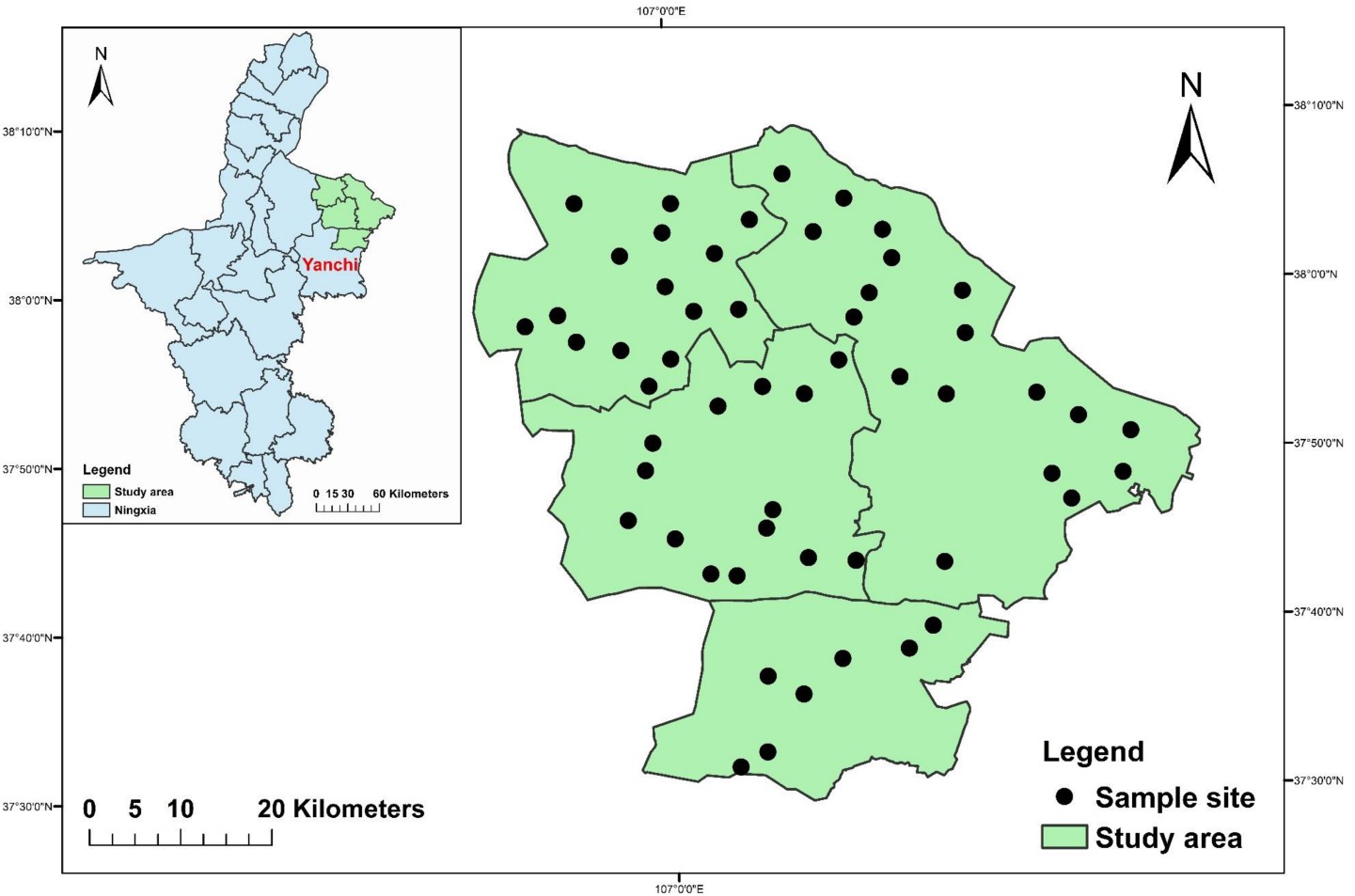

**Figure 1 Study area and sample sites.**

**Table 1 Vegetation profiles of the sample plots.**

| Degradation degree | Symbol | Height/ cm | Coverage/ % | Density/ (plant·m$^{-2}$) | Aboveground biomass/ (g·m$^{-2}$) | Species composition |
|---|---|---|---|---|---|---|
| Undegraded | UD | 15.22 | 57.38 | 87.29 | 53.83 | 12–14 species in Gramineae and Asteraceae, 4–6 species in Leguminosae, Chenopodiaceae, etc., species distributed evenly |
| Lightly degraded | LD | 12.19 | 45.22 | 72.37 | 46.17 | 9–11 species in Gramineae and Asteraceae, 5–7 species in Leguminosae, Chenopodiaceae, Convolvulaceae, etc., species distributed relatively evenly |
| Moderately degraded | MD | 18.25 | 20.61 | 41.51 | 29.11 | 6–7 species in Gramineae and Asteraceae, 5–6 species in Leguminosae, Chenopodiaceae, Euphorbiaceae, etc., species distributed unevenly |
| Severely degraded | SD | 11.56 | 12.31 | 15.33 | 18.25 | 2–3 species in Gramineae and Asteraceae, 5–7 species in Leguminosae, Chenopodiaceae, Asclepiadaceae, Zygophyllaceae, etc., more annual plants and a large proportion of poisonous weeds, species distributed unevenly |

were randomly measured and used to calculate its average height. In each quadrat of the sample plot, we measured the coverage and the density of vegetation, and then pruned each species to the ground. The plants were brought back to the laboratory to obtain the aboveground biomass by drying at 65 °C to a constant weight.

(ii) Soil moisture content, soil bulk density determination, and soil sample collection: Across two consecutive years, we used the TDR probe to measure the soil moisture content of the 0–5, 5–15, and 15–30 cm soil layers during the middle of each month from April to October, and took the average of each month as the soil moisture content data. The ring knife method (*Liu & Jiang, 1996*) was used to measure the soil bulk density of the 0–5, 5–15, and 15–30 cm soil layers in August once a year across 2 years. The soil samples of the 0–5, 5–15, and 15–30 cm soil layers were also collected in August once a year across two years to determine other soil physical properties, chemical properties, microorganisms, and enzyme activities. The soil samples collected to determine microorganisms and enzyme activities were placed into a 4 °C incubator. Bring the soil samples back to the laboratory for subsequent processing.

(iii) Determining soil physical and chemical properties (*Liu & Jiang, 1996*) and microorganism and enzyme activities (*Yao & Huang, 2006*): The soil particle composition was measured using a Microtrac S3500 laser particle size analyzer. The soil organic matter content was determined using the potassium dichromate volumetric method. The soil total nitrogen content was determined using the fully automatic Kjeldahl method. The soil available potassium content was determined using the flame photometer method. The soil available phosphorus content was determined using the Olsen method. The number of soil microorganisms was determined using the plate coating culture counting method. Beef extract peptone agar medium was used for aerobic bacteria, Martin medium was used for fungi, and modified Gao's No. 1 medium was used for actinomycetes. Three dilution gradients ($10^{-3}$, $10^{-4}$, and $10^{-5}$) were used for each sample and three replicates were used at each dilution gradient. Soil urease activity was determined using indophenol colorimetry, polyphenol oxidase activity was determined using spectrophotometry, protease activity was determined using the Folin phenol method, phosphatase activity was determined using the phenyl disodium phosphate colorimetric method, and sucrase activity was determined using the 3,5-dinitrosalicylic acid colorimetric method.

### Data analysis

We analyzed each depth interval separately at different degradation degrees. The experimental data were statistically processed using Excel 2016 and then analyzed using SPSS26. The soil indicators were subjected to a one-way ANOVA and the correlation analysis was conducted based on a Pearson correlation. Principal component analysis (PCA) was performed simultaneously.

### Soil quality evaluation

There were many soil physical, chemical, and biological indicators, and while they showed differences, they also exhibited certain correlations. To more directly reflect the

comprehensive soil quality situation in the desert steppe at different degrees of degradation, we transformed multiple relevant soil factors into a few comprehensive factors using PCA. Indicators with high factor loading were selected to establish the MDS, and the SQI was used to evaluate the overall soil quality of grasslands at different degrees of degradation.

Based on the concept of dimensionality reduction (*Bilgili, Kucuk & Van Es, 2017*), PCA transforms multiple related original variables into several comprehensive indicators under the premise of ensuring less information is lost. When performing PCA, we first processed the soil indicators and made them dimensionless, and the membership value of each indicator was obtained (*Jahany & Rezapour, 2020*; *Nabiollahi et al., 2017*). Then, we used PCA for the soil indicators and the standard of "the eigenvalue >1" to determine the number of principal components (*Juhos et al., 2019*). The accumulated variance contribution rates should reach 85%, and the eigenvalue, variance contribution rate, and the factor score coefficient matrix of each principal component were obtained. The weight of each indicator was calculated using the ratio of the common factor variance of each indicator to the sum of the common factor variances of all indicators obtained by PCA (*Askari & Holden, 2014*).

The MDS, used to evaluate soil quality, was proposed by Larson & Pierce in 1991 (*Zhao et al., 2019*). The factor loading of each indicator in each principal component was obtained using PCA. It is generally believed that the greater the absolute value of the factor loading, the greater the weight of the variable in the corresponding principal component, and the positive and negative only represent the influence effect. It is assumed that soil variables with high factor loading can best represent soil quality variation characteristics (*Nabiollahi et al., 2018*). Therefore, the MDS only retains high load attributes in each factor. The high load attribute is defined as an absolute value within 10% of the highest factor load (*Nosrati & Collins, 2019*). When multiple attributes are retained in a single factor, a multivariate correlation coefficient is used to determine whether the variable will be considered redundant, and subsequently eliminated, from the MDS (*Rahmanipour et al., 2014*). Variables with good correlation are considered redundant, so only one can be considered for the MDS. If variables with high weight are not related, each variable is considered important and a variable in the MDS (*Raiesi, 2017*).

The SQI is the integration of soil quality evaluation indicators, with a range 0–1 (*Lima et al., 2013*). The larger the SQI, the better the soil quality. The formula for calculating the SQI is:

$$\text{SQI} = \sum_{i=1}^{n} Q(x_i) \cdot W_i \tag{1}$$

where $Q(x_i)$ is the membership value of soil indicator $i$, $W_i$ is the weight of soil indicator $i$, and $n$ is the number of soil indicators (*Jahany & Rezapour, 2020*).

**Table 2 The main soil physical and chemical properties of grassland at different degrees of degradation (average ± standard error).**

| Soil layer/cm | Degradation degree | <0.05 mm soil clay/% | Soil bulk density/ (g·cm⁻³) | Soil moisture/ % | Organic matter/ (g·kg⁻¹) | Total nitrogen/ (g·kg⁻¹) | Available potassium/ (mg·kg⁻¹) | Available phosphorus/ (mg·kg⁻¹) |
|---|---|---|---|---|---|---|---|---|
| 0–5 | UD | 4.82 ± 0.46aA | 1.47 ± 0.03aA | 10.27 ± 1.23aA | 4.34 ± 0.32aA | 0.63 ± 0.07aA | 140.00 ± 13.89aA | 4.30 ± 0.42bA |
| | LD | 4.56 ± 0.77aA | 1.52 ± 0.01aA | 9.09 ± 0.59aA | 3.65 ± 0.22aA | 0.61 ± 0.04aA | 129.40 ± 5.18aA | 7.25 ± 0.68abA |
| | MD | 3.45 ± 1.86aA | 1.55 ± 0.04aA | 8.12 ± 0.75aA | 3.58 ± 0.51aA | 0.59 ± 0.10aA | 125.00 ± 9.57aA | 6.05 ± 3.31abA |
| | SD | 3.05 ± 0.90aA | 1.56 ± 0.03aA | 6.79 ± 2.13aA | 3.26 ± 0.12aA | 0.47 ± 0.05aAB | 68.42 ± 3.27bA | 8.31 ± 0.59aA |
| 5–15 | UD | 6.56 ± 0.98aA | 1.51 ± 0.03aA | 10.02 ± 2.28aA | 4.31 ± 0.45aA | 0.72 ± 0.05aA | 112.50 ± 16.56aA | 3.40 ± 0.20bAB |
| | LD | 5.25 ± 0.93aA | 1.53 ± 0.02aA | 8.43 ± 0.66aA | 3.10 ± 0.19bAB | 0.62 ± 0.04abA | 108.51 ± 5.41aB | 5.35 ± 0.47abB |
| | MD | 4.14 ± 1.90aA | 1.57 ± 0.06aA | 6.37 ± 0.92aA | 2.62 ± 0.52bA | 0.50 ± 0.05bA | 106.25 ± 1.75aB | 8.04 ± 2.65aA |
| | SD | 3.40 ± 1.91aA | 1.57 ± 0.03aA | 6.17 ± 0.03aA | 2.30 ± 0.49bAB | 0.50 ± 0.03abA | 62.14 ± 3.42bA | 6.91 ± 0.99aAB |
| 15–30 | UD | 5.01 ± 1.45aA | 1.56 ± 0.03aA | 7.75 ± 1.49aA | 2.91 ± 0.38aB | 0.77 ± 0.08aA | 101.25 ± 13.81aA | 2.45 ± 0.52aB |
| | LD | 3.63 ± 0.69aA | 1.56 ± 0.01aA | 7.44 ± 0.62aA | 2.58 ± 0.26aB | 0.66 ± 0.04aA | 76.23 ± 4.62aC | 4.33 ± 0.51aB |
| | MD | 3.54 ± 0.53aA | 1.59 ± 0.04aA | 6.42 ± 0.62aA | 2.39 ± 0.47aA | 0.59 ± 0.08abA | 74.50 ± 2.22abC | 5.12 ± 1.65aA |
| | SD | 2.44 ± 1.00aA | 1.63 ± 0.05aA | 5.83 ± 0.61aA | 2.03 ± 0.31aB | 0.38 ± 0.03bB | 48.56 ± 2.17bB | 5.22 ± 0.54aB |

**Note:**
Different lowercase letters (a, b) after the same column of data indicate significant differences ($p < 0.05$) between different degrees of degradation; different capital letters (A, B) after the same column of data indicate significant differences ($p < 0.05$) between different soil layers.

## RESULTS

### Changes in the physical and chemical soil properties of grassland at different degrees of degradation

The main physical and chemical soil properties are shown in Table 2. In each layer of the soil that experienced grassland degradation, the soil clay content and moisture content both decreased, the soil bulk density increased, indicating that the soil had become more compacted as a result of degradation, and the soil organic matter content showed an overall downward trend (in the 5–15 cm soil layer, it decreased significantly (F = 3.64, $p = 0.017$)), and the soil total nitrogen content showed an overall downward trend (in the 15–30 cm soil layer, it decreased significantly (F = 4.75, $p = 0.004$)). With grassland degradation, the soil available potassium content significantly decreased across all soil layers ($p < 0.05$), especially in severely degraded grassland, and the change in the soil available phosphorus content only showed a significant different in the 5–15 cm soil layer (F = 3.29, $p = 0.026$).

As the soil depth increased, there was no obvious change in the soil clay content. The soil bulk density increased, indicating that the deeper the soil layer, the more compacted the soil. The soil moisture content showed an overall downward trend. In grassland at different degrees of degradation, the soil organic matter content decreased as soil depth increased. There was a significant difference ($p < 0.05$) for grassland at all degrees of degradation except for the moderately degraded grassland. As soil depth increased, the soil total nitrogen content showed an upward trend in the undegraded and lightly degraded grassland; the soil available potassium content decreased in all grasslands at different degrees of degradation (the difference was significant ($p < 0.05$) except for the undegraded grassland); and the soil available phosphorus content decreased in the

undegraded, lightly degraded, and severely degraded grassland (the difference was significant ($p < 0.05$) for the undegraded and lightly degraded grassland).

## Changes in the soil microorganism quantity of grassland at different degrees of degradation

The changes in the quantity of soil microorganisms in grassland at different degrees of degradation are shown in Fig. 2. In the horizontal distribution pattern, the number of microorganisms (bacteria, actinomycetes, and fungi) decreased with grassland degradation, showing the following ranking: undegraded >lightly degraded >moderately degraded >severely degraded. The undegraded and lightly degraded grassland had the greatest number of bacteria, followed by the number of actinomycetes, and then the number of fungi ($p < 0.05$). In the moderately degraded and severely degraded grassland, the difference in the quantity of microorganisms was not significant ($p > 0.05$). In terms of the vertical distribution pattern, the number of bacteria, actinomycetes, and fungi in the undegraded, lightly degraded, and moderately degraded grassland changed in the same order (upper soil >middle soil >lower soil), and there was a significant difference ($p < 0.05$) between the soil layers. In the severely degraded grassland, the quantity of microorganisms in each soil layer was extremely small, and there was no significant difference ($p > 0.05$) between the soil layers.

## Changes in the soil enzyme activity in grassland at different degrees of degradation

The changes in the soil enzyme activity in grassland at different degrees of degradation are shown in Fig. 3. Soil enzyme (urease, polyphenol oxidase, protease, phosphatase, and sucrase) activity showed a general downward trend with grassland degradation. Soil phosphatase activity decreased significantly ($p < 0.05$) with degradation; the activity of urease, polyphenol oxidase, protease, and sucrase was significantly different ($p < 0.05$) among the undegraded, moderately degraded, and severely degraded grassland; and the activity of urease and polyphenol oxidase was significantly different ($p < 0.05$) between the lightly degraded and severely degraded grassland. In terms of soil layer distribution, phosphatase activity decreased significantly in the deeper soil layers of the undegraded and lightly degraded grassland; the activity of urease, polyphenol oxidase, and sucrase in the upper layer of the undegraded grassland was significantly higher than in the middle and lower layers, and there was no significant difference in the other degraded grassland; and protease activity was only significantly higher in the upper layer of the lightly degraded grassland compared to the middle and lower layers.

## Correlation analysis between soil factors

The results of the correlation analysis between the soil factors are shown in Table 3. There were some close correlations among the soil factors. Soil bulk density was significantly negatively correlated with soil clay content, moisture content, organic matter content, total

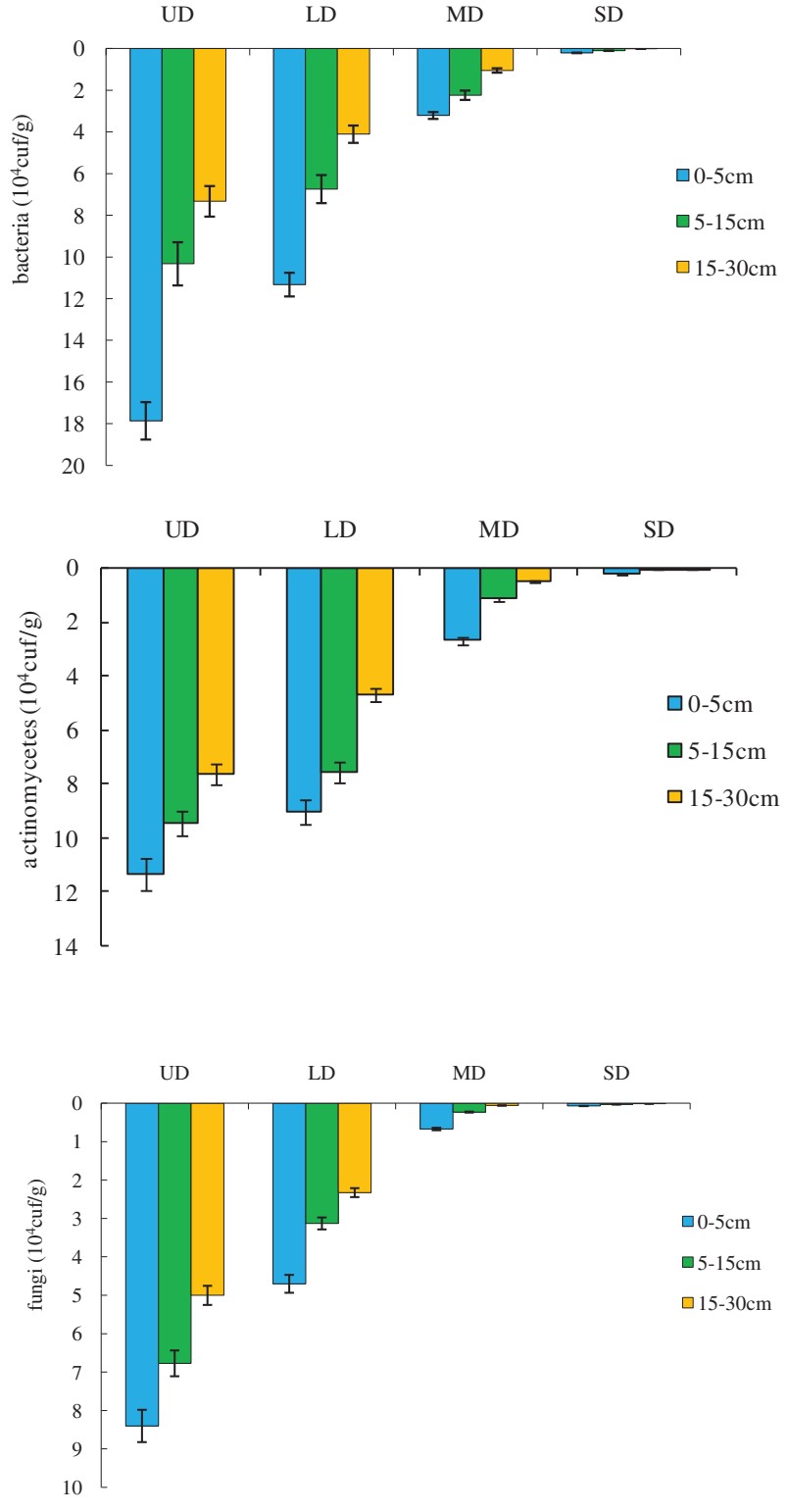

**Figure 2 Soil microorganism quantity in grassland at different degrees of degradation.**

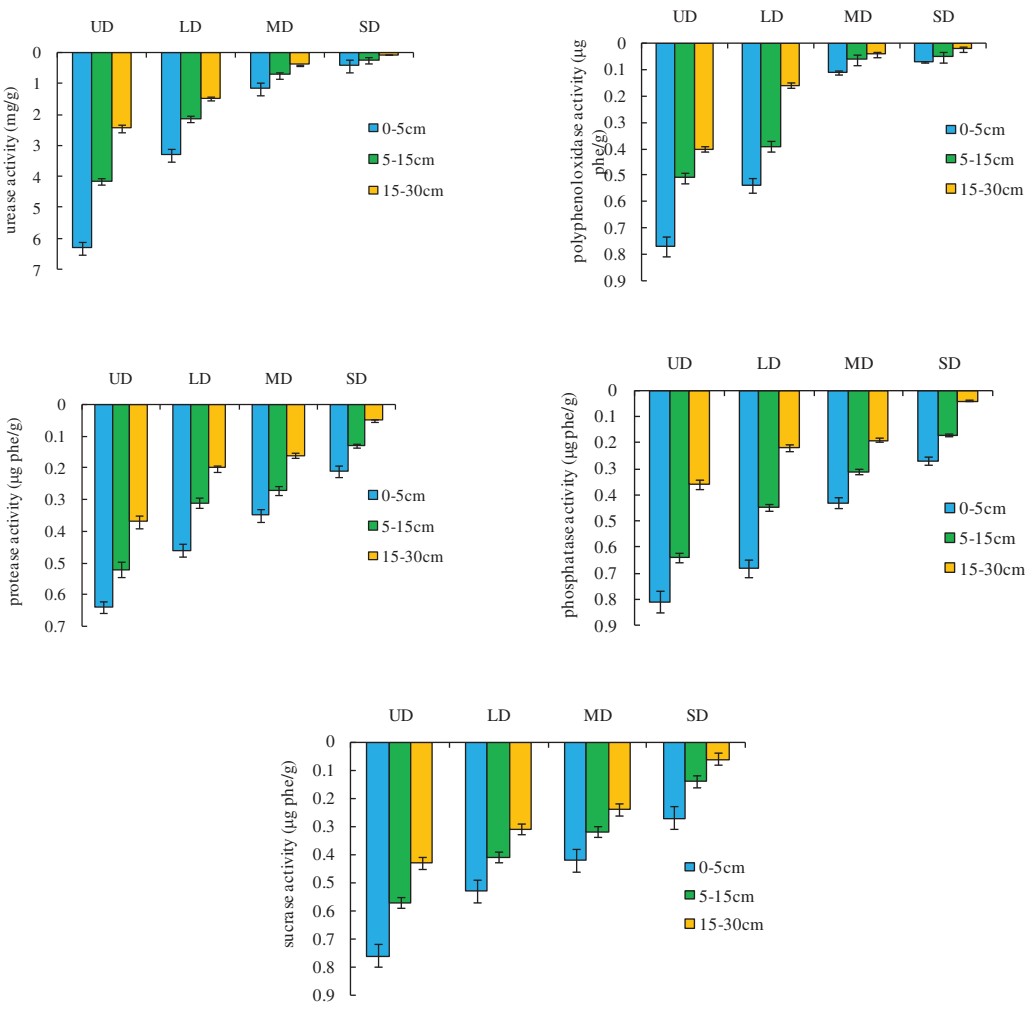

**Figure 3 Soil enzyme activity in grassland at different degrees of degradation.**

nitrogen content, available potassium content, the activity of the five enzymes, and microorganism quantity. Soil clay content and moisture content were positively related to each other, and both were positively related to soil organic matter content, total nitrogen content, and available potassium content. The soil organic matter content and total nitrogen content were significantly positively correlated with soil available potassium content, while the soil total nitrogen content was significantly negatively related to the soil available phosphorus content. The soil clay content, moisture content, organic matter content, total nitrogen content, and available potassium content were all significantly positively correlated with the activity of the five enzymes and microorganism quantity. The soil available phosphorus content was significantly negatively related to the fungi quantity, and there was a significantly positive correlation between the activity of the five soil enzymes and the microorganism quantity.

Ma et al. (2022), PeerJ, DOI 10.7717/peerj.13100

Peerj

**Table 3 The correlation between soil factors.**

| | <0.05 mm soil clay | Soil bulk density | Soil moisture | Organic matter | Total nitrogen | Available potassium | Available phosphorus | Urease activity | Polyphenol oxidase activity | Protease activity | Phosphatase activity | Sucrase activity | Bacteria | Actinomycetes |
|---|---|---|---|---|---|---|---|---|---|---|---|---|---|---|
| Soil bulk density | −0.72** | | | | | | | | | | | | | |
| Soil moisture | 0.78** | −0.93** | | | | | | | | | | | | |
| Organic matter | 0.68* | −0.90** | 0.93** | | | | | | | | | | | |
| Total nitrogen | 0.78** | −0.59* | 0.67* | 0.52 | | | | | | | | | | |
| Available potassium | 0.66* | −0.83** | 0.83** | 0.81** | 0.57* | | | | | | | | | |
| Available phosphorus | −0.49 | 0.19 | −0.41 | −0.19 | −0.71** | −0.14 | | | | | | | | |
| Urease activity | 0.73** | −0.90** | 0.94** | 0.85** | 0.62* | 0.78** | −0.46 | | | | | | | |
| Polyphenol oxidase activity | 0.76** | −0.88** | 0.92** | 0.80** | 0.64* | 0.77** | −0.45 | 0.97** | | | | | | |
| Protease activity | 0.77** | −0.93** | 0.95** | 0.93** | 0.65* | 0.91** | −0.32 | 0.94** | 0.92** | | | | | |
| Phosphatase activity | 0.73** | −0.95** | 0.95** | 0.93** | 0.56* | 0.91** | −0.19 | 0.92** | 0.91** | 0.98** | | | | |
| Sucrase activity | 0.75** | −0.94** | 0.95** | 0.91** | 0.67* | 0.91** | −0.34 | 0.94** | 0.91** | 0.99** | 0.97** | | | |
| Bacteria | 0.71** | −0.89** | 0.93** | 0.81** | 0.62* | 0.81** | −0.43 | 0.99** | 0.98** | 0.94** | 0.93** | 0.94** | | |
| Actinomycetes | 0.82** | −0.85** | 0.93** | 0.77** | 0.76** | 0.77** | −0.54 | 0.94** | 0.97** | 0.89** | 0.88** | 0.90** | 0.95** | |
| Fungi | 0.80** | −0.84** | 0.91** | 0.79** | 0.72** | 0.70** | −0.59* | 0.97** | 0.97** | 0.90** | 0.86** | 0.89** | 0.96** | 0.97** |

**Notes:**
* Significant correlation at the 0.05 level.
** Extremely significant correlation at the 0.01 level.

**Table 4 Principal component eigenvalue, variance contribution rate, and factor score coefficient matrix.**

| Evaluation indicator | Principal component | |
|---|---|---|
| | 1 | 2 |
| Soil clay ($X_1$) | −0.0150 | 0.1786 |
| Soil bulk density ($X_2$) | 0.1385 | −0.1032 |
| Soil moisture ($X_3$) | 0.0862 | 0.0064 |
| Organic matter ($X_4$) | 0.1539 | −0.1402 |
| Total nitrogen ($X_5$) | −0.1124 | 0.3488 |
| Available potassium ($X_6$) | 0.1537 | −0.1452 |
| Available phosphorus ($X_7$) | 0.2350 | −0.5379 |
| Urease activity ($X_8$) | 0.0701 | 0.0365 |
| Polyphenol oxidase activity ($X_9$) | 0.0613 | 0.0525 |
| Protease activity ($X_{10}$) | 0.1185 | −0.0562 |
| Phosphatase activity ($X_{11}$) | 0.1558 | −0.1323 |
| Sucrase activity ($X_{12}$) | 0.1129 | −0.0452 |
| Bacteria ($X_{13}$) | 0.0770 | 0.0225 |
| Actinomycetes ($X_{14}$) | 0.0203 | 0.1325 |
| Fungi ($X_{15}$) | 0.0100 | 0.1513 |
| Characteristic root | 12.2490 | 1.4521 |
| Variance contribution rates/% | 81.6601 | 9.6808 |
| Accumulated variance contribution rates/% | 81.6601 | 91.3409 |

## Soil quality evaluation

### Soil factor PCA

The PCA results are shown in Table 4. According to the PCA, the eigenvalues of the first two principal components were all greater than 1 (12.25 and 1.45, respectively), and their accumulated variance contribution rate reached 91.34%. This indicated that the first two principal components could represent the original 15 soil indicators and accurately reflect the quality of the soil.

### Selection of the minimum data set

Principal component 1 had several high-weight variables, namely, the soil organic matter content, soil available potassium content, soil available phosphorus content, and phosphatase activity. Since the soil organic matter content was significantly related to the soil available potassium content and the phosphatase activity, we excluded the soil available potassium content with the smallest absolute value of factor loading. The variables entered into the MDS were the soil organic matter content, soil available phosphorus content, and phosphatase activity. Principal component 2 had several high-weight variables, namely, the soil total nitrogen content and soil available phosphorus content. Since the soil available phosphorus content was already entered into the MDS, we selected the soil total nitrogen content to enter into the MDS at this time. In summary, the

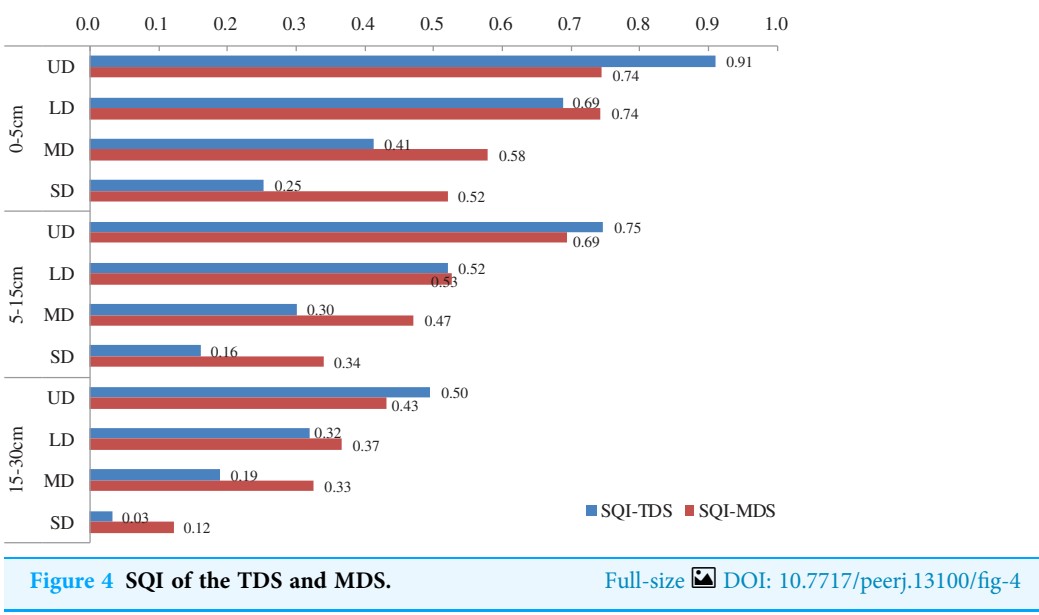

**Figure 4 SQI of the TDS and MDS.**

soil indicators in the MDS were the soil organic matter content, soil total nitrogen content, soil available phosphorus content, and phosphatase activity.

### Calculating the SQI

Using the membership values and weight coefficients of the four indicators in the MDS, we calculated the SQI values of the grassland at different degradation degrees. The results showed that the SQI values of the undegraded, lightly degraded, moderately degraded, and severely degraded grassland were 0.62, 0.55, 0.46, and 0.33, respectively. The overall soil quality was ranked as undegraded grassland >lightly degraded grassland >moderately degraded grassland >severely degraded grassland.

### Rationality verification of the MDS

The SQI of the TDS was called the SQI-TDS, and the SQI of the MDS was called the SQI-MDS. We calculated the SQI-TDS and SQI-MDS of the grassland separately at different degradation degrees for each depth interval (Fig. 4). A scatter plot of the SQI-TDS and SQI-MDS was made for regression analysis (Fig. 5). According to the fitting effect, SQI-TDS was highly correlated with SQI-MDS, and the $R^2$ value was 0.814. This proved that the MDS could replace the TDS when evaluating the soil quality of the desert steppe at different degradation degrees.

## DISCUSSION

### Changes in soil physical and chemical characteristics in the desert steppe at different degrees of degradation

Grassland degradation leads to long-term biodiversity loss, instability (*Raiesi & Salek-Gilani, 2020*), and impacts on soil physical and chemical properties in grassland ecosystems. Plant coverage decreases, which leads to an increase in soil bulk density and a decrease in soil structure stability, the soil clay content shows a decreasing trend, soil wind

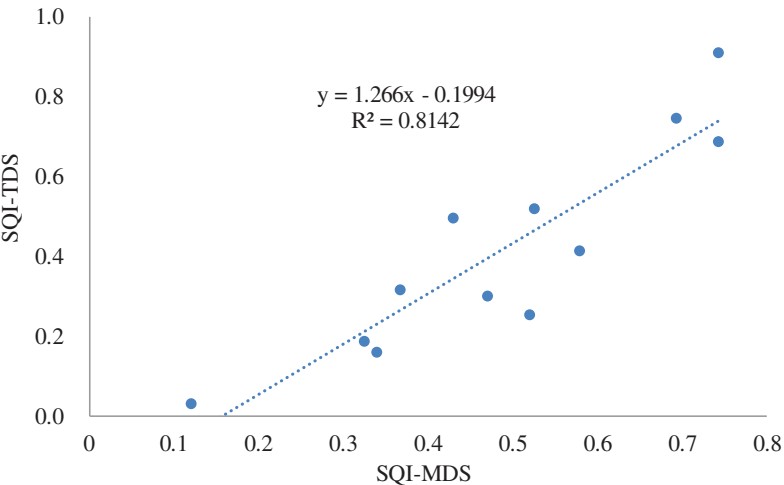

**Figure 5 Linear relationship between the SQI values of the TDS and MDS.**

erosion increases, the amount of soil clay particles blown away by the wind increases, and the soil water and fertilizer conservation capacity decreases, resulting in soil degradation. The decrease in soil moisture content in degraded grassland is also related to poor soil water holding capacity because soil clay content is directly related to soil water and fertilizer conservation capacity. As soil depth increases, there is no obvious change in the soil clay content, which might be due to the relatively stable structure of soil particles and a lag in soil changes across the deeper layers. The combination of grassland degradation and the increase in soil depth caused the soil bulk density to increase, which was consistent with the results of *Yao et al. (2016)* and *Peng et al. (2018)* and indicated that grassland degradation is significantly related to increased soil compaction. Soil compaction reduced the soil pore volume and destroyed the macropores responsible for the majority of gas and water movements, resulting in decreased air and water in the soil. Grassland degradation led to a decrease in soil moisture content, which was consistent with the results of *Yuan et al. (2019)* and *Yang et al. (2019)*. Grassland degradation increased the evaporation rate and surface runoff and reduced the soil water infiltration capacity (*Tiscornia, Jaurena & Baethgen, 2019*). As soil depth increased, the soil moisture content decreased, which was related to the lack of precipitation in the desert steppe of arid windy sandy areas.

Due to the low availability of rainfall and water in arid and semiarid areas, the organic input of plant biomass and the content of soil organic matter were low (*Raiesi & Salek-Gilani, 2020*). With simultaneous grassland degradation and the increase in soil depth, soil organic matter content showed a downward trend. This was because the decrease in plant biomass and coverage led to a decrease in the carbon content in the organic carbon pool (*Zhang et al., 2019*). Severely degraded grassland had poor vegetation coverage, and there was no continuous participation of organic matter. Therefore, the decline in soil organic matter was one of the most important indicators of grassland degradation (*Yaşar Korkanç & Korkanç, 2016*). As soil depth increased, the decrease in plant litter

accumulation and the amount of roots resulted in the decline of soil organic matter content and the deterioration of soil properties. Soil total nitrogen content also showed a downward trend with grassland degradation. This was due to the carbon, nitrogen, and phosphorus elements in grassland soil being mainly from organic matter, and the changing soil texture, decreasing vegetation coverage, and changing vegetation composition were all considered reasons for the decline in soil total nitrogen content with grassland degradation (*Dong et al., 2012*). As soil depth increased, the soil total nitrogen content showed an upward trend in the undegraded and lightly degraded grassland, which might have been related to the nutrient loss in the surface soil.

The changes in the soil available potassium and available phosphorus might have been due to the changes in the soil microorganism composition and the enzyme activity affecting the mineralization of the total nutrients. Changes in the aboveground vegetation affected the absorption of the available nutrients by plants.

## Soil biological characteristic changes of the desert steppe at different degrees of degradation

The number of microorganisms showed a downward trend with grassland degradation. These differences were attributed to the differences in vegetation characteristics and soil properties of the grassland at different degrees of degradation. Research by *Li et al. (2016)* found that soil nutrient status was the most important factor controlling the composition changes of bacteria and fungi. Grassland degradation damaged plant growth, soil structure, and nutrient status. The disappearance of aboveground plants limited the development and population of soil microbial communities (*Zhou et al., 2019*), and the decrease in litter input in the soil led to a decrease in the availability of microbial substrates (*Wu et al., 2014*). Additionally, decreasing soil moisture content, decreasing soil nutrients, and increasing soil compaction were not beneficial to the growth and reproduction of microorganisms. The composition proportions of the three kinds of microorganisms were different in the grassland at different degrees of degradation, which might have been related to the different biological attributes of the microbial communities. Bacteria are small and reproduce faster than actinomycetes and fungi. Previous studies found that in nutrient-deficient soil, fungi were less dependent on *in situ* nutrients than bacteria (*Sarathchandra et al., 2005*). In terms of vertical distribution, the distribution of soil microorganisms was closely related to soil physical structure (*Li et al., 2016*), soil nutrients, and soil respiration. In the undegraded, lightly degraded, and moderately degraded grassland, the upper soil had the largest number of microorganisms, which was consistent with previous research (*Sanaullah et al., 2016*). This is because the upper soil contained various kinds of litter spoilage rich in organic matter, and had a loose and porous structure that provided relatively good living conditions for microorganisms. Moreover, the temperature of the upper soil was higher than that of the lower soil, making the degradation of plant litter greater and the soil enzyme activity higher (*van Bruggen & Semenov, 2000*), which were all helpful for the growth and reproduction of various microorganisms (*Pandey, Agrawal & Bohra, 2015*). In the severely degraded grassland, the vegetation was sparse, the surface soil was exposed to the air, soil desertification was

serious, and the temperature of the surface soil changed greatly. The environment was not conducive to the growth of soil microorganisms, resulting in a small number of microorganisms and the distribution of microorganisms moving down the soil profile. Therefore, the vertical difference was not significant.

Soil enzyme activity is a potential indicator of soil quality. It characterizes the biological activity of soil and participates in biochemical functions and nutrient cycling (*Maurya et al., 2020*). It can quickly respond to microenvironmental changes in the soil (*Pandey, Agrawal & Bohra, 2015*), is highly sensitive to external interference, and is easy to measure. Soil enzyme activity showed a downward trend with grassland degradation because soil enzyme activity is closely related to soil organic matter content and soil physical and chemical properties, and the soil nutrient poorness during the process of grassland degradation caused the decline of enzyme activity. The decrease in enzyme activity was also related to the decrease in microbial biomass and the change in microbial composition in degraded soil because the enzymes involved in carbon, nitrogen, phosphorus, and sulfur mineralization were mainly from microorganisms (*Raiesi & Beheshti, 2015*). The soil enzyme activity decreased with the increasing soil depth, which was consistent with the results of *Raiesi & Beheshti (2015)* and *Sanaullah et al. (2016)*. This was because as soil depth increased, soil organic matter decreased, and soil temperature and moisture content decreased, which limited the ability of soil microorganisms to metabolize and produce enzymes.

## Correlations between soil factors of the desert steppe at different degrees of degradation

According to correlation analysis, we found that soil factors were closely related. As soil bulk density increased, soil compactness increased, resulting in a reduction of air and water in the soil, which had a large impact on the soil physical and chemical properties and soil microorganisms and enzymes (*Peng et al., 2018*; *Yao et al., 2016*). *Cookson, Murphy & Roper (2008)* found that an increase in soil carbon storage was directly related to soil clay content. *Xu et al. (2019)* found that with grassland degradation, the reduction of clay content led to a decrease in carbon and nitrogen storage capacity, and the decrease in soil moisture content reduced microbial activity, resulting in a decrease in the mineralization rate and, subsequently, in available nitrogen and phosphorus concentrations. *Yang et al. (2019)* and *Maurya et al. (2020)* found that soil moisture content strongly affected soil physical and chemical properties and microbial diversity. Soil nutrients were an important factor affecting the composition and quantity of microorganisms (*Li et al., 2016*). Soil organic matter is the fundamental source of soil microbial nutrients, and the large majority of organic matter is also the final product of microbial metabolism (*Cookson, Murphy & Roper, 2008*). *Pan et al. (2013)* found that soil enzyme activity was significantly positively correlated with soil moisture content, organic carbon, and total nitrogen concentration. *Yang et al. (2018)* and *Zhang et al. (2020)* found that soil organic carbon had the greatest impact on soil enzyme activity. There was a significantly positive correlation between the activity of the five soil enzymes and the amount of the three kinds of microorganisms, indicating that microorganism quantity and enzyme activity both

promoted and restricted each other (*Raiesi & Beheshti, 2015*). All these results proved that soil physical, chemical, and biological properties are closely related and work together to determine soil quality.

## Soil quality evaluation of the desert steppe at different degrees of degradation

Determining soil quality integrates various soil properties that are dynamic and sensitive to external environmental changes. Soil quality cannot be obtained by directly measuring grassland degradation. It needs to be evaluated by combining mathematical logic of certain physical, chemical, and biological characteristics (*Li et al., 2019*). The application of multivariate statistical methods can be used to make the data meaningful (*Obade & Lal, 2014*). When evaluating soil quality, it is impossible to measure using only one single feature or all features of the soil (*Maurya et al., 2020*). The most important variables are selected to construct a MDS. To evaluate the soil quality of degraded desert steppe, the MDS was comprised of soil organic matter content, soil total nitrogen content, soil available phosphorus content, and phosphatase activity. Soil organic matter, a direct product of the biological activities of plants, animals, and many other biological factors that affect soil functions, affects the physical, chemical, and microbiological properties of the soil and the availability of nutrients (*Obade & Lal, 2014*; *Raiesi, 2017*; *Sharma et al., 2016*). Total nitrogen is a major nutrient for vegetation growth and an important indicator used to measure the level of soil fertility (*Pham, Nguyen & Kappas, 2018*). Soil available phosphorus is an important limiting factor that affects vegetation growth and an important indicator used to evaluate the level of soil phosphorus supply. Phosphatase activity can directly affect the decomposition and transformation of soil organic phosphorus and its biological effectiveness. The four indicators are closely related to soil fertility and vegetation growth and are important indicators for the soil quality evaluation of degraded desert steppe. The MDS evaluation results were consistent with the results of the TDS, which proved that the MDS was able to replace the TDS when evaluating the soil quality of the desert steppe at different degradation degrees.

The desert steppe in arid windy sandy areas has an extremely fragile ecosystem, and most grasslands are degraded to different degrees. The soil organic matter content is low, soil wind erosion is serious, and the mass reproduction of weeds (poisonous and non-poisonous) sacrifice high-quality species (*Liu & Diamond, 2005*). As the plant community structure degraded, plant height and coverage decreased, plant flora was simplified, and the biomes decreased (*Lu, Cong & Li, 2017*). Changes in plant species composition and coverage changed litter input, root structure, and soil physical, chemical, and biological properties (*Guo et al., 2018*; *Wang & Fu, 2020*; *Wang et al., 2014*). Vegetation degradation promoted soil degradation, and soil degradation caused vegetation changes, forming a strong feedback mechanism. With grassland degradation, the changes in soil microorganism quantity and enzyme activity were not completely consistent. This result showed that in the desert steppe's vegetation-soil changing process, changes in some soil properties lagged behind changes in vegetation. The changes in soil physical, chemical, and biological properties at different degrees of degradation were different, reflecting the

complexity of changes in soil factors. However, the correlation between grassland soil factors showed that soil physical, chemical, and biological properties were connected, interacted with each other, and affected and determined the direction of grassland soil quality evolution together. The comprehensive SQI illustrated this point. The overall soil quality order based on SQI values was undegraded grassland >lightly degraded grassland >moderately degraded grassland >severely degraded grassland, indicating that the degradation of the desert steppe in arid windy sandy areas had relatively consistent effects on the physical, chemical, and biological traits of the soil.

Long-term positioning research can be used to determine more systematic ecological grassland changes. However, since establishing permanent research plots is limited by many factors, it is difficult to measure ecological changes of the same plot across a time series. Therefore, the space for time substitution method is generally recognized by scholars when studying ecological succession (*Liang et al., 2002*; *Shen et al., 2015*; *Zhao et al., 2020*). In this study, we used the method to select sample plots with close distances and consistent topography and soil conditions in order to ensure the consistency of the initial conditions of the sample plots as much as possible.

## CONCLUSION

As grassland degraded, the soil bulk density increased; soil clay, moisture, organic matter, total nitrogen, and available potassium content decreased; and soil microorganism quantity and enzyme activity decreased. As the soil depth increased, the soil bulk density increased; soil moisture, organic matter, available potassium, and available phosphorus content decreased; and soil microorganisms accumulated in the upper soil of undegraded, lightly degraded, and moderately degraded grassland. There was a positive correlation among the soil clay content, moisture content, organic matter content, total nitrogen content, available potassium content, microorganism quantity, and enzyme activity, while the soil bulk density was negatively correlated with the above factors. The MDS used for soil quality evaluation of degraded desert steppe was comprised of soil organic matter content, soil total nitrogen content, soil available phosphorus content, and phosphatase activity. Based on the MDS, the SQI values of undegraded, lightly degraded, moderately degraded, and severely degraded desert steppe in arid windy sandy areas were 0.62, 0.55, 0.46, and 0.33, respectively, and the overall order of soil quality was undegraded grassland >lightly degraded grassland >moderately degraded grassland >severely degraded grassland. The results showed that the degradation of the desert steppe in arid windy sandy areas had relatively consistent effects on the physical, chemical, and biological traits of the soil. Vegetation degradation led to soil degradation, but soil degradation lagged behind vegetation degradation. The MDS can be used to replace the TDS when evaluating the soil quality of the desert steppe at different degrees of degradation.

## ACKNOWLEDGEMENTS

We would like to thank Qiaoneng Hu, Huan Kou, and Jiawei Niu for their help during the field survey and sampling collection. We are indebted to the editors and reviewers of this paper for their constructive comments and suggestions.

### Funding

This work was supported by the National Natural Science Foundation of China (No.32060406), the Ningxia Science and Technology Innovation Leader Training Project (No.KJT2018003), the Ningxia Natural Science Foundation Project (No.2021AAC03019), the Ningxia Key Research and Development Program (No. 2020BBF02003), the First-class Discipline Construction Project (Grassland Science Discipline) for the high school in Ningxia (No. NXYLXK2017A01), and the Science and Technology Innovation Guidance Project of Autonomous Region Agricultural Science and Technology Independent Innovation Special (No. NKYJ-20-11). The funders had no role in study design, data collection and analysis, decision to publish, or preparation of the manuscript.

### Grant Disclosures

The following grant information was disclosed by the authors:
National Natural Science Foundation of China: 32060406.
Ningxia Science and Technology Innovation Leader Training Project: KJT2018003.
Ningxia Natural Science Foundation Project: 2021AAC03019.
Ningxia Key Research and Development Program: 2020BBF02003.
First-class Discipline Construction Project (Grassland Science Discipline) for the high school in Ningxia: NXYLXK2017A01.
Science and Technology Innovation Guidance Project of Autonomous Region Agricultural Science and Technology Independent Innovation Special: NKYJ-20-11.

### Competing Interests

The authors declare that they have no competing interests.

### Author Contributions

- Jing Ma performed the experiments, analyzed the data, prepared figures and/or tables, authored drafts of the paper, and approved the final draft.
- Jianrong Qin conceived and designed the experiments, performed the experiments, analyzed the data, prepared figures and/or tables, and approved the final draft.
- Hongbin Ma conceived and designed the experiments, authored or reviewed drafts of the paper, and approved the final draft.
- Yao Zhou performed the experiments, analyzed the data, prepared figures and/or tables, and approved the final draft.
- Yan Shen conceived and designed the experiments, authored or reviewed drafts of the paper, and approved the final draft.
- Yingzhong Xie conceived and designed the experiments, authored or reviewed drafts of the paper, project administration, and approved the final draft.
- Dongmei Xu conceived and designed the experiments, authored or reviewed drafts of the paper, and approved the final draft.

## Data Availability

The raw measurements are available in the Supplemental File.

## Supplemental Information

Supplemental information for this article can be found online at http://dx.doi.org/10.7717/peerj.13100#supplemental-information.

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
