# Peer review of "Soil characteristic changes and quality evaluation of degraded desert steppe in arid windy sandy areas"

_PeerJ, doi:10.7717/peerj.13100_

## Round 0.1 · original submission · Major Revisions

Both reviewers i think have given insightful comments which i recommend the authors follow closely. I agree some clarity of expression is required broadly throughout the manuscript and much could be done to remove repetition which ultimately muddles the work a lot. Some thorough editing and chopping of irrelevant and repeated text will see a great improvement in this pretty good manuscript. Additionally:

Line 196-198. Soil moisture is described as measured every month for 2 year. What was the sampling and measurement frequency for the other variables? Together these data were measured at different depths and to my understanding different frequencies, but there is very little in the methods description how these data were handled for the subsequent analyses for as the PCA and SQI work.

Each of the described soil measurement methods is based on some published methodology but citation of these methods is not provided.
Need to describe more about the NIR method for estimating available P and K. In general, if the NIR method is based on a model then some description of the model is needed and how skilful it is. Generally, if the NIR approach is via diffuse reflectance then i find it difficult to believe that good estimates of available nutrients would be possible with this method.

The input variables into the PCA work need some better description. From my understanding there is a whole time series of soil moisture data, and a collection of soil and vegetation data collected perhaps once only or maybe over several campaigns of the couple of year. First clarify this uncertainty and then talk about how this data was treated prior to the PCA work.

Need to be explicit in the methods that the analysis was conducted for each depth interval separately, not all together.

Line 389-394. What is the significance of these expressions and what can one do with them? I understand each of the Xs is a soil parameter, but you will need to define each of them explicitly, or perhaps just refer to the associated table.

Figure 5. What is the source of the that this plot is produced? There is not a lot of data here compared to the number of study sites.

Reviewer 1 ·

Basic reporting

The quality of the research presented is limited by an inconsistent ability to communicate the main ideas of the paper. Further editing is required to ensure that the paper reads as intended and communicates appropriately the methodologies undertaken and the results identified. Current issues that need revision include:
The concept of the TDS is not adequately explored on its introduction in line 125, but a good summary is provided later in 256.
The confusion of the terms index, indicator and attribute which are used somewhat interchangeably within the methods to refer to a property of the soil that may be included in the TDS/MDS, which is particularly problematic at line 231. Effort should be made to use distinct terms to refer to a variable before and after calculating the membership values in the distribution function, with the use of the term index for the transformed values on different scales being particularly confusing.
The methods for PCA and the transformations involved are reiterated in section 3.5.1 of your results, 370-382. This can largely be removed, but much of it is clearer than your original description and you may wish to use much of the wording here in the methods.
You incorrectly refer to microbial counts increasing with land degradation in 319, should be decreased.

Beyond this, there are many examples of unclear or awkward sentences which confuse the point of the paper, including at lines: 57, 59, 76, 92, 202, 228, 283, 550, 563 and across more locations. The current phrasing makes comprehension difficult. I suggest you have a colleague who is proficient in English and familiar with the subject matter review your manuscript.

Experimental design

The experimental design seems sufficient for a study of this nature, however i would advise that care be taken in explaining how the substitution of space for time in your analysis addresses the practical constraints of sampling in line 174.

Validity of the findings

Your findings are relatively well addressed within the discussion of the paper, and i thank you for the inclusion of the data set analysed. However I feel the paper is lacking in its discussion of the variables identified as contributing towards the MDS for degradation assessment. You explore the relationship between many variables and degraded landscapes within section 4.3, but a stronger link to the specific MDS subset in the discussion and conclusions would benefit the paper.

Additional comments

Overall the paper is an interesting exploration of the impacts of grassland degradation on soil quality, with sound methodology. Fault lies primarily in the clarity of the writing, with further editing and revision of the language used required before the paper is suitable for acceptance.

Reviewer 2 ·

Basic reporting

This study keeps using repeated sentences in the manuscript. Please remove the duplicated findings in difference paragraphs. The whole manuscript is too long as well and makes the reader fail to grasp the key information.
I would like to suggest restructuring the manuscript as a lot of information was delivered in the Results section, but they were supposed to be in the Materials & Methods section, for example, the weight of each indicator (Wi).
The grey area in Figure 1 was not explained.
Table 4 seems unnecessary. It is enough to just keep the accumulated variance of 2 principal components.

Experimental design

no comment

Validity of the findings

It seems hard to get the key point due to the meaningless duplicated information.
I would like to suggest selecting some key findings to describe not saying all of them.

Additional comments

What is “characteristic root฀1” at Line 246
line 389-394 unnecessary to show such information
line 400-402 This information was duplicated. It was mentioned before but once again here

---

## Round 0.2 · Minor Revisions

This paper is getting close. Please respond to the reviewer suggestions which will not take very long and then it should be right to go for publication.

Reviewer 2 ·

Basic reporting

The current version is much clearer than the first one, with sufficient data, analysis and comparisons based on literature references.
At line 86 it is better to add references for the methods you mentioned.

Experimental design

The experimental design is good especially involving soil biological properties for soil quality evaluation.
The research questions have been well discussed.

Validity of the findings

The findings of MDS, TDS and SQI are consistent with ground truth and with other studies.

Additional comments

The study is almost ready to be accepted except the following minor issues.
Line 73 Monitoring soil quality is necessary to assess parameter changes.... Please clarify what parameters?
Line 95 dot(.) should be comma (,)
Line 211 no need to say as cited
Line 301,302 and other places. Please reduce the digits of the those numbers to be 2.
Line 418 remove sharply.

---

## Round 0.3 · accepted · Accept

Thank you for sharing this research and your efforts through the review process to see it through to publication. Well-done!